# Adverse outcomes in patients hospitalized with pneumonia at age 60 or more: A prospective multi-centric hospital-based study in India

Suman Kanungo[1]*, Uttaran Bhattacharjee[1], Aslesh O. Prabhakaran[2], Rakesh Kumar[3], Prabu Rajkumar[4], Sumit Dutt Bhardwaj[5], Alok Kumar Chakrabarti[1], Girish Kumar C. P.[4], Varsha Potdar[5], Byomkesh Manna[1], Ritvik Amarchand[3], Avinash Choudekar[6], Giridara Gopal[3], Krishna Sarda[1], Kathryn E. Lafond[7], Eduardo Azziz-Baumgartner[7], Siddhartha Saha[2], Lalit Dar[6], Anand Krishnan[3]

**1** ICMR-National Institute of Cholera and Enteric Diseases, Kolkata, India, **2** Influenza program, US Centers for Disease Control and Prevention, New Delhi, India, **3** Centre for Community Medicine, All India Institute of Medical Sciences, New Delhi, India, **4** ICMR-National Institute of Epidemiology, Chennai, India, **5** ICMR-National Institute of Virology, Pune, India, **6** Department of Microbiology, All India Institute of Medical Sciences, New Delhi, India, **7** Influenza Division, US Centers for Disease Control and Prevention, Atlanta, Georgia, United States of America

* sumankanungo@gmail.com

**Data Availability Statement:** All relevant data are within the manuscript and its Supporting Information files.

## Abstract

### Background

Limited data exists regarding risk factors for adverse outcomes in older adults hospitalized with Community-Acquired Pneumonia (CAP) in low- and middle-income countries such as India. This multisite study aimed to assess outcomes and associated risk factors among adults aged ≥60 years hospitalized with pneumonia.

### Methods

Between December 2018 and March 2020, we enrolled ≥60-year-old adults admitted within 48 hours for CAP treatment across 16 public and private facilities in four sites. Clinical data and nasal/oropharyngeal specimens were collected by trained nurses and tested for influenza, respiratory syncytial virus (RSV), and other respiratory viruses (ORV) using the qPCR. Participants were evaluated regularly until discharge, as well as on the 7th and 30th days post-discharge. Outcomes included ICU admission and in-hospital or 30-day post-discharge mortality. A hierarchical framework for multivariable logistic regression and Cox proportional hazard models identified risk factors (e.g., demographics, clinical features, etiologic agents) associated with critical care or death.

### Findings

Of 1,090 CAP patients, the median age was 69 years; 38.4% were female. Influenza viruses were detected in 12.3%, RSV in 2.2%, and ORV in 6.3% of participants. Critical care was required for 39.4%, with 9.9% in-hospital mortality and 5% 30-day post-discharge mortality.

**Funding:** The study was funded by the Centers for Disease Control and Prevention, Atlanta, USA under Co-operative Agreement U01IP001074. The funders had no role in study design, data collection and analysis, decision to publish, or preparation of the manuscript.

**Competing interests:** All the authors have declared that no competing interests exist.

Only 41% of influenza CAP patients received antiviral treatment. Admission factors independently associated with ICU admission included respiratory rate >30/min, blood urea nitrogen>19mg/dl, altered sensorium, anemia, oxygen saturation <90%, prior cardiovascular diseases, chronic respiratory diseases, and private hospital admission. Diabetes, anemia, low oxygen saturation at admission, ICU admission, and mechanical ventilation were associated with 30-day mortality.

## Conclusion

High ICU admission and 30-day mortality rates were observed among older adults with pneumonia, with a significant proportion linked to influenza and RSV infections. Comprehensive guidelines for CAP prevention and management in older adults are needed, especially with the co-circulation of SARS-CoV-2.

## Introduction

Although lower respiratory tract infections cause millions of deaths every year, which disproportionately occur among older adults in low- and middle-income countries, there is insufficient information about their etiology and associated risk factors for adverse outcomes to inform policy for prevention and management in such settings [1]. For example, an estimated 2.74 million deaths have occurred globally from lower respiratory tract infections (LRI) in 2015, with a 3.2% rise in fatalities between 2005 and 2015 [2]. Pneumonia, an acute lower respiratory tract infection, is a common morbidity across all age groups, especially among older adults. Older age is associated with a 10-fold increased risk of community acquired pneumonia and an increased rate of hospitalization, as well as adverse outcomes [3, 4]. Globally it is estimated pneumonia resulted in 6.8 million hospitalizations and 1.1 million deaths among adults aged ≥65 years in 2015 [5]. Similarly, in another review, globally there were an estimated 2.8 million influenza associated hospitalizations among adults aged ≥ 65 years [6].

The majority of the evidence about community acquired pneumonia (CAP) associated hospitalization burden and outcomes are from high-income countries. Mortality rates among patients in these countries ranged from 8 to 14% and up to 50% among those requiring intensive care unit (ICU) admission [7, 8]. CAP hospitalization and mortality in middle income countries, however, can be dramatically higher [9, 10]. Despite limited data from African and Asian regions, a review identified CAP as one of the main reasons for adult hospitalizations in these regions [11]. In a retrospective hospital-based study in Thailand, 23% of the patients with CAP were admitted to ICU and 30-day mortality rate was 19% [12].

In middle-income countries like India, there are few studies about CAP-associated hospitalization among older adults which focused on clinical features or etiologic agents; rarely do these describe the outcomes in terms of ICU admission or 30-day mortality [13–16]. Information about etiology and factors related to outcome would help to inform policy about the potential value of vaccination to prevent CAP, triaging, and clinical management among older adults aged ≥ 60 years. We conducted this study to estimate the proportion of adverse outcomes among older adults (aged ≥60 years) admitted with CAP at multiple centers, including intensive care unit admission and mortality in hospital or within 30-day post discharge, and identify clinical, epidemiological, and viral etiological factors associated with adverse outcomes.

## Materials and methods

### Study setting

This multi-centric hospital-based study was a part of the 'Indian Network of Population-Based Surveillance Platforms for Influenza and Other Respiratory Viruses among the Elderly" (INSPIRE) and has been described earlier [17]. The enrolled hospitals in Delhi were coordinated by the All-India Institute of Medical Sciences (New Delhi, India), Kolkata by National Institute of Cholera and Enteric Diseases, Pune by National Institute of Virology (Pune, India) and in Chennai by National Institute of Epidemiology (Chennai, India). At each site, four hospitals—two public (one each secondary and tertiary) and two private (one secondary and one tertiary) were selected based on number of admissions of older adults with acute respiratory illnesses (Fig 1) and operational convenience. Secondary care facilities were those with less than 100 beds with or without ICU facility whereas tertiary hospitals were to have more than 100 beds with ICU facility.

### Enrolments of cases

All patients aged ≥60 years admitted in medicine, pulmonary and geriatric wards of the selected facilities between 4th December 2018 to 20th March 2020 were screened using an operational definition modified from 2009 update of British Thoracic Society (BTS) guidelines, described as new onset cough within last 10 days along with at least one lower respiratory tract symptom (dyspnea, chest pain), and at least one systemic feature (sweating, fevers, shivers, aches and pains and/or temperature of 38˚C or more) and tachypnea with a respiratory rate> 20/min or a physician diagnosed pneumonia [7]. Patients who had been hospitalized for > 48 hours in the hospital or any other facility for the same episode of illness were not enrolled to exclude possible nosocomial illnesses.

At the time of enrollment, trained project nurses collected data about age, sex, influenza vaccination in past one year, history of current smoking and use of alcohol and pre-existing chronic morbidity by interviewing participants. Details of clinical assessment, diagnostic tests results of hemogram, blood urea, blood glucose, X-ray and use of medications were gathered by reviewing medical records using a standard data collection tool on open data kit (ODK) modules on handheld tablets.

The study team visited the hospitals every alternate day to enroll new patients and evaluate the existing ones till discharge. Any change in status of the patient or admission to the ICU

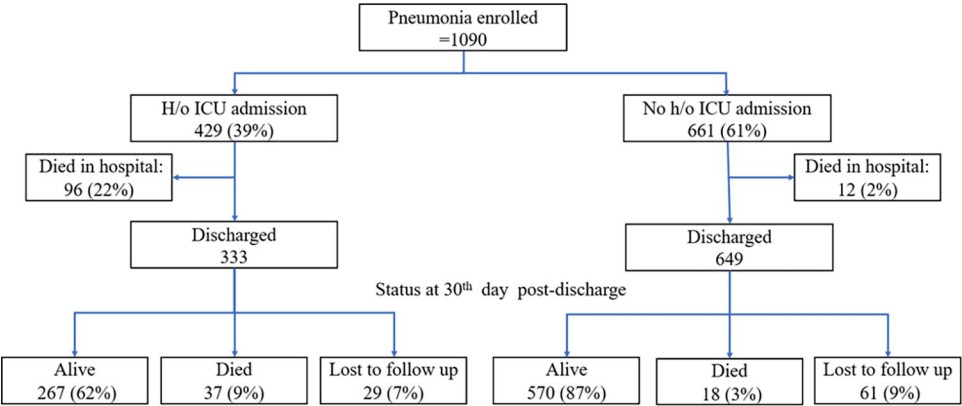

**Fig 1. Flowchart of enrolled hospitalized community acquired pneumonia patients aged > = 60 years and their outcomes in four sites in India, 2018–2020.**

was noted. After discharge, participants were followed up telephonically on day 7 and day 30 from the date of discharge to collect data on their outcome status. During telephonic follow-up, if a participant could not be contacted on first attempt, two more attempts were made to contact within next 7 days, before categorizing a participant as lost to follow-up. The outcomes were categorized as alive and at home, alive and readmitted, dead, or lost to follow-up.

## Laboratory testing for influenza and RSV

Nasal and oropharyngeal specimens were collected from enrolled participants within 48 hours of admission using the standard protocol. These combined swabs were placed immediately into viral transport media (VTM) on ice or ice pack, triple-sealed for transportation. All the specimens were transported to the respective virology laboratory of the institute on the same day. Viral RNA was extracted using the QiAmp Viral RNA kit (Qiagen, Germany) as per the manufacturer's protocols. We used U.S. Centers for Disease Control and Prevention (CDC) approved one-step Real Time-PCR protocol and primers and probes for influenza and RSV virus to detect the influenza (A and B) and RSV virus. The primers and probes were provided by CDC International Resource Reagent (https://www.internationalreagentresource.org). All samples were also tested for human metapneumovirus (hMPV), parainfluenza viruses (PIV), rhinoviruses and adenoviruses using a multiplex kit developed by National institute of Virology [18].

## Statistical analysis

Data was collected in android-based questionnaire developed in Open Data Kit (ODK) and imported into STATA-16 version (StataCorp LLC) for statistical analysis. Two main adverse clinical outcomes were considered–admission in ICU care and mortality. Participants needing ICU care were those who were either admitted to ICU or referred to other facility for ICU admission. Mortality was recorded as deaths during hospitalization or within 30 days after hospital discharge. The proportion of participants with adverse clinical outcome were compared using chi-squared test by age groups, sex, viral RNA detection, self-reported co-morbid conditions, type of hospital (e.g., private vs. public), clinical and laboratory parameters at the time of admission. We categorized the clinical and laboratory parameters as described in pneumonia severity index, CURB-65 scoring (CURB-65 score evaluate the clinical severity of CAP and the acronym of "CURB-65" stands for C: **C**onfusion or mental status altered; U: increased **U**rea (Blood Urea Nitrogen) level; R: high **R**espiratory Rate; B: low **B**lood Pressure, 65: age $\geq$**65**), and WHO classification of anaemia [19–21]. To identify factors associated with ICU admission we estimated adjusted odds ratio (adj OR) and their 95% confidence intervals using multivariable logistic regression with a hierarchical framework. We also applied the same hierarchical framework to identify factors associated with mortality by estimating hazard ratio (HR) using multivariable Cox proportional hazard regression models. For the hierarchical framework we categorized the independent variables into four levels (Table 1). The most distal level in hierarchal framework or level-1 (L1) included baseline characteristics of participants like age group, sex, smoking status, comorbid conditions (chronic respiratory disease, cardiovascular disease including hypertension, diabetes). Level 2 (L2) variables included etiological agents (influenza or RSV). Level 3 (L3) included clinical and laboratory condition at the time of admission including mental status (altered sensorium vs conscious & oriented), high respiratory rate (> = 30 per minute), low blood pressure (Systolic blood pressure <90mmHg or Diastolic BP< = 60mm Hg), low oxygen saturation (<90% without oxygen), anemia (hemoglobin level <12gm/dl for females and <13gm/dl for males), total leucocyte count (<5000/dl or >20,000/dl), elevated blood urea nitrogen (>19mg/dl). The most proximal or

**Table 1. Factors associated with intensive care unit admission among older adults hospitalized with pneumonia in India 2018–2020.**

| Factors | Model 1 Adj. odds ratio (95% CI) | Model 2 Adj. odds ratio (95% CI) | Model 3 Adj. odds ratio (95% CI) | Model 4 Adj. odds ratio (95% CI) | Model 5 Adj. odds ratio (95% CI) | Model 6 Adj. odds ratio (95% CI) | Model 7 Adj. odds ratio (95% CI) |
|---|---|---|---|---|---|---|---|
| Age group | | | | | | | |
| 65–74 year | 0.83 (0.61–1.13) | - | - | - | 0.84 (0.62–1.15) | 0.89 (0.59–1.35) | 0.89 (0.57–1.37) |
| > = 75 year | 1.42 (1.01–2.00) | - | - | - | 1.43 (1.02–2.01) | 1.26 (0.79–2.01) | 1.46 (0.88–2.41) |
| Chronic Respiratory Disease | 1.61 (1.25–2.08) | - | - | - | 1.61(1.24–2.08) | 1.57 (1.11–2.22) | 1.57 (1.08–2.27) |
| Cardiovascular disease or hypertension | 2.02(1.55–2.63) | - | - | - | 1.99 (1.53–2.59) | 1.67(1.17–2.22) | 1.84 (1.25–2.70) |
| Diabetes | 1.42 (1.07–1.89) | - | - | - | 1.40 (1.05–1.87) | 1.34 (0.91–1.98) | 1.51 (1.00–2.31) |
| Current smoking history | 0.66 (0.47–0.91) | - | - | - | 0.68 (0.49–0.93) | 0.76 (0.49–1.17) | 0.85 (0.53–1.35) |
| Influenza | - | 1.39 (0.96–1.99) | - | - | 1.32 (0.91–1.94) | 0.98 (0.57–1.68) | 0.96 (0.54–1.72) |
| Respiratory rate > = 30 per min | - | - | 3.56 (1.65–7.68) | - | - | 3.33 (1.73–6.39) | 3.71 (1.77–7.77) |
| Blood urea nitrogen >19gm/dl | - | - | 2.15 (1.39–3.33) | - | - | 1.55 (1.09–2.20) | 1.63 (1.11–2.39) |
| Altered sensorium | - | - | 33.56(11.05–101.87) | - | - | 8.71 (4.12–18.39) | 15.87 (6.85–36.74) |
| Anemia | - | - | 1.73 (1.08–2.78) | - | - | 1.55 (1.06–2.28) | 1.74 (1.17–2.60) |
| Oxygen saturation | | | | | | | |
| 90–93% | - | - | 2.9 (1.66–5.05) | - | - | 3.29 (2.08–5.19) | 2.87 (1.76–4.66) |
| <90% | - | - | 8.67 (4.79–15.67) | - | - | 9.88 (6.27–15.56) | 7.3 (4.46–11.98) |
| Private sector hospitalization | - | - | - | 8.72 (6.3–12.19) | - | - | 8.23 (5.10–13.29) |

Model 1 include level 1 factors (age group, sex, cardiovascular disease, diabetes, chronic respiratory disease, current smoking status); Model 2 include level 2 factors (Influenza, RSV);Model 3 included level 3 factors (Mental Status, respiratory rate, body temperature, oxygen saturation, blood urea nitrogen, total leukocyte count, anemia);Model 4 included type of hospital; Model 5 included age-group, cardio vascular diseases, chronic respiratory disease, diabetes, current smoking and influenza; Model 6 included age-group, cardio vascular diseases, chronic respiratory disease, diabetes, current smoking, influenza, respiratory rate, blood urea nitrogen level, mental status, anemia and oxygen saturation; Model 7 included age-group, cardiovascular diseases, chronic respiratory disease, diabetes, current smoking, influenza, respiratory rate, blood urea nitrogen level, mental status, anemia and oxygen saturation and type of hospital. Anemia- Hemoglobin > = 12gm/dl in female & > = 13 in male.

Level 4 (L4) variables included type of care received including ICU admission, mechanical ventilation, oseltamivir therapy and type of hospital (i.e., private vs. public). Regression was conducted using variables at each level (model 1–4) separately to identify variables with p value<0.1. Subsequently, regression was conducted with selected L1, L2 variables (p value <0.1) (model 5); L1, L2, L3 variable (model 6); and L1, L2, L3, L4 variable (model 7) (Table 1).

## Ethics

The study was reviewed and approved by the institutional review board [IRB]/institutional ethics committee [IEC] of all the participating institutions (AIIMS IEC Ref No.: AIIMS-IEC-283/02.06.17, PR 12/2017; ICMR-NIE IEC Ref No.: NIE-IHEC/2017-03; ICMR-NIV IEC Ref No.: NIV-IEC/2018/D-5; ICMR-NICED IEC Ref No.: NICED-A-1/2017-IEC) and the Health Ministry's Screening Committee (HMSC) of India. All the study institutes obtained necessary administrative permission from selected hospitals for enrolling eligible patients and accessing the hospital data. Written informed consent of the participants or family caregivers was obtained before enrolment into this study.

## Results

We screened 18,769 older adults admitted in the selected hospitals at four sites using operational definition of CAP to find 1,151 (6.1%) eligible cases of which 1,090 consented and provided nasal & oropharyngeal specimen. Among the enrolled cases, 399 (36.6%) were from public hospitals and 691 (63.4%) were from private hospitals (Table 2). The median age of participants was 69 years (IQR: 63–75) and 419 (38.0%) were female. Overall, 43.0% of

**Table 2. Characteristics and outcomes of enrolled older adults (> = 60 years) hospitalized with pneumonia in four sites in India (Dec 2018 –Mar 2020).**

| Variables | Delhi n (%) | Chennai n (%) | Kolkata n (%) | Pune n (%) | All sites n (%) |
|---|---|---|---|---|---|
| No. of pneumonia cases | 424 | 247 | 267 | 152 | 1090 |
| **Type of facility** | | | | | |
| Public secondary | 30 (7.1) | 12 (4.9) | 17 (6.4) | 15 (9.9) | 74 (6.8) |
| Public tertiary | 87 (20.5) | 133 (53.8) | 55 (20.6) | 50 (32.9) | 325 (29.8) |
| Private secondary | 7 (1.7) | 15 (6.1) | 13 (4.9) | 3 (2.0) | 38 (3.5) |
| Private tertiary | 300 (70.8) | 87 (35.2) | 182 (68.2) | 84 (55.3) | 653 (59.9) |
| **Age group** | | | | | |
| 60–64 years | 131 (30.9) | 73 (29.6) | 68 (25.5) | 41 (27.0) | 313 (28.7) |
| 65–74 years | 174 (41.4) | 128 (51.8) | 117 (43.8) | 67 (44.1) | 486 (44.6) |
| ≥75 years | 119 (28.1) | 46 (18.6) | 82 (30.7) | 44 (28.9) | 291 (26.7) |
| Females | 167 (39.4) | 82 (33.2) | 85 (31.8) | 85 (55.9) | 419 (38.4) |
| Health Insurance coverage | 221 (52.1) | 57 (23.1) | 162 (60.7) | 17 (11.2) | 457 (41.9) |
| Current Smokers | 76 (17.9) | 93 (37.7) | 54 (20.2) | 11 (7.2) | 234 (21.5) |
| **Co-morbidities** | | | | | |
| Diabetes | 89 (21.0) | 71 (28.7) | 95 (35.6) | 50 (32.9) | 305 (28.0) |
| Cardiovascular diseases including hypertension | 146 (30.4) | 85 (34.4) | 144 (53.9) | 121 (79.6) | 496 (45.5) |
| Chronic Respiratory Disease | 198 (46.7) | 91 (36.8) | 162 (60.7) | 97 (63.8) | 548 (50.3) |
| Ever had tuberculosis | 39 (9.2) | 12 (4.9) | 8 (3) | 0 | 66 (6.1) |
| Any other comorbidity | 65 (15.3) | 15(6.1) | 26(9.7) | 20 (13.2) | 126 (11.6) |
| **Viruses detected** | | | | | |
| Influenza virus | 40 (9.4) | 29 (11.7) | 45 (16.9) | 20 (13.2) | 134 (12.3) |
| A(H1N1)pdm09 | 18 (4.2) | 7 (2.8) | 13 (4.9) | 8 (5.3) | 46 (4.1) |
| A(H3N2) | 14 (3.3) | 13 (5.3) | 30 (11.2) | 7 (4.6) | 64 (5.9) |
| B (Yamagata) | 0 (0) | 4 (1.6) | 0 (0) | 0 (0) | 4 (0.4) |
| B (Victoria) | 8 (1.9) | 5 (2) | 2 (0.7) | 5 (3.3) | 20 (1.8) |
| RSV | 8 (1.9) | 8 (3.2) | 4 (1.5) | 4 (2.6) | 24 (2.2) |
| Adenovirus | 2 (0.8) | 3 (0.7) | 1 (0.4) | 2 (1.3) | 8 (0.7) |
| Para influenza virus (1–4) | 5 (2) | 4 (0.9) | 8 (3) | 5 (3.3) | 22 (2) |
| Rhinovirus | 5 (2) | 5 (1.2) | 2 (0.7) | 4 (2.6) | 16 (1.5) |
| Human metapneumovirus | 7 (2.8) | 2 (0.5) | 2 (0.7) | 12 (7.9) | 23 (2.1) |
| **Care received** | | | | | |
| Antibiotic | 423 (99.8) | 219 (88.7) | 252 (94.4) | 149 (98) | 1044 (95.8) |
| Antiviral | 35 (8.3) | 6 (2.4) | 109 (40.8) | 99 (65.1) | 249 (22.8) |
| ICU admission | 133 (31.4) | 20 (8.1) | 195 (73.0) | 81 (53.3) | 429 (39.4) |
| Put on mechanical ventilator | 6 (1.4) | 8 (3.2) | 27 (10.1) | 51 (33.6) | 92 (8.4) |
| **Outcome** | | | | | |
| Died in hospital | 10 (2.4) | 5 (2.0) | 72 (27.0) | 21 (13.8) | 108 (9.9) |
| Died within 30-day post discharge | 13 (3) | 5 (2) | 36 (13.5) | 1 (0.7) | 55 (5) |
| Lost to follow-up | 35 (8.3) | 8 (3.2) | 38 (14.2) | 9 (5.9) | 90 (8.3) |

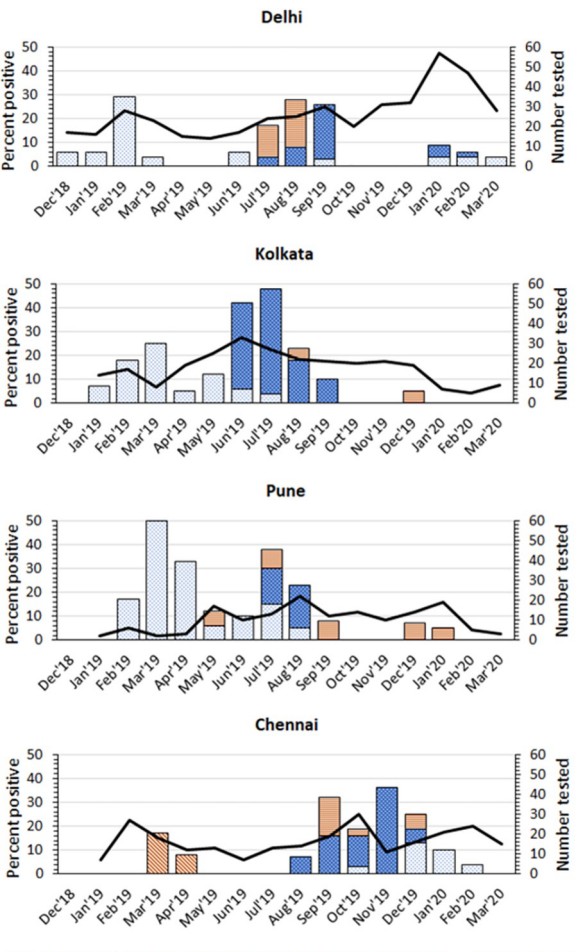

**Fig 2. Monthly influenza positivity (%) among hospitalized pneumonia in-patients aged 60 years and above in four sites in India, 2018–2020.**

participants had insurance coverage; 39.4% of females and 45.7% of males (p value 0.04). The main co-morbidities reported were chronic respiratory disease (CRD) (50.3%), cardiovascular diseases (CVD) including hypertension (44.9%) and diabetes (28.0%). About one in five participants (21.5%) were current smokers. Only 8 (0.7%) participants reported having received influenza vaccine within the past year.

Influenza was detected in 134 (12.3%) cases and of these, 47.8% were A(H3N2) and 34.3% were A(H1N1)pdm09 (Table 2). In Delhi, Kolkata and Pune, percent of specimens testing positive for influenza peaked during June-September, while in Chennai influenza peaked during September-December months (Fig 2). A secondary peak was noted in February -March of 2019 at all sites. Fifty percent of influenza cases in Delhi, 60% in Pune and 76% in Kolkata occurred during June- September, while in Chennai, 71% of influenza cases occurred between September to December. RSV was detected in 24 (2.2%), hMPV in 23 (2.1%), PIV in 22 (2%), Rhinovirus in 16 (1.5%) and adenovirus in 8 (0.7%) pneumonia cases.

The data needed for calculating the CURB-65 score at admission were available from 50% of participants. Among these, 8% were classified as severe pneumonia (CURB-65 score > 2): 21.7% in Pune, 8.7% in Kolkata, 4.7% in Delhi and 2.9% Chennai. The percent of participants

**Table 3. Comparison of outcomes by type of facilities and virus detected among older adults (> = 60 years) hospitalized with pneumonia in four sites in India (Dec 2018 –Mar 2020).**

| Outcomes n (%) | Etiologic agent detected | | | | | Type of hospital | | |
|---|---|---|---|---|---|---|---|---|
| | All Subjects | Non-Influenza, Non-RSV | Influenza | RSV* | p value | Private | Public | p value |
| N | 1090 | 934 | 134 | 22 | | 691 | 399 | |
| ICU admission | 429 (39.4%) | 356 (38.1%) | 62 (46.2%) | 11 (50%) | 0.11 | 380 (55%) | 49 (12.3%) | < .00001 |
| Mechanical Ventilation | 92 (8.4%) | 74 (7.9%) | 15 (11.2%) | 3 (13.6%) | 0.3 | 72 (10.4%) | 20 (5%) | < .002 |
| Mortality | 163 (14.9%) | 130 (13.9%) | 29 (21.6%) | 4 (18.1%) | 0.058 | 120 (17.4%) | 43 (10.5%) | < .004 |

with CURB-65 score > 2 admitted in private sector hospitals was 8.4%, compared 7.1% in public hospitals (p value 0.6%).

A total of 249 (22.8%) participants had received antiviral (oseltamivir) while 1044 (95.8%) had received antimicrobials. Antiviral use was observed in 41 (16.7%) out of 245 CAP admitted within 48 hours of symptom onset as compared to 208 (24.6%) out of 845 CAP admitted after 48 hours (p value 0.01). Among the influenza positives, 133 (99%) were treated with antibiotics and 56 (41%) were treated with antivirals. A total of 429 (39.4%) pneumonia cases needed ICU care (range 8.1% in Chennai to 73% in Kolkata). Of all pneumonia cases, 108 (9.9%) died during hospitalization and another 55 (5%) were known to have died within 30 days of discharge (Fig 1). Ninety (8.3%) patients were loss to follow up after discharge even after three attempts to contact them telephonically. In-hospital mortality was highest in Kolkata (27%) and lowest in Chennai (2%) and Delhi (2.4%). The high ICU admission and mortality rate occurred in Kolkata where there was one private tertiary facility with ICU admission rates of 97.8% and mortality of 51.6%.

We found requirement for ICU care was similar in influenza associated pneumonia (46.2%), RSV associated pneumonia (50%) and non-influenza non-RSV associated pneumonia cases (38.1%) (p value 0.11) (Table 3). In-hospital or post-discharge 30-day mortality rate was 21.6% for influenza associated pneumonia compared to 13.6% of non-influenza-non-RSV pneumonia cases (p value 0.058). In our study, ICU use (55% Vs 12%, p value<0.01) and mortality (17.5% Vs 10.4%, p value<0.01) among those admitted in private hospital were higher than in public hospital (Table 3).

Univariate analysis showed that ICU admissions were observed significantly more (p value <0.05) among participants aged ≥75 years (49.1%), with comorbid conditions like CRD (43.9%), diabetes (48.5%), and CVD (50.7%). Significant factors (p value<0.05) at admission associated with requirement of ICU care were altered sensorium (83.1%), respiratory rate≥30/min (71.2%), body temperature >102°F (68.2%), oxygen saturation <90% (73.9%), anemia (44.1%) or blood urea nitrogen >19gm/dl (54.8%). Hierarchical multivariable analysis suggest that CRD (adj OR 1.57; 95% CI 1.08–2.27), CVD (adj OR 1.84; 95% CI 1.25–2.70), respiratory rate > = 30 per min (adj OR 3.71; 95%CI 1.77–7.77), elevated blood urea nitrogen (adj OR 1.63; 95% CI 1.11–2.39), altered sensorium (adj OR 15.87; 95%CI 6.85–36.74), anemia (adj OR 1.74; 95%CI 1.17–2.60), oxygen saturation <90% (OR 7.3, 95%CI 4.46–11.98) and private hospital admission (adj OR 8.23; 95% CI 5.10–13.29) were significantly associated with need for ICU care (Table 1).

Mortality rates in-hospital or within 30-day post discharge were significantly higher (p value <0.05) among those aged ≥75 years (20.9%), and with comorbidities like diabetes (23%) and CVD (20.7%). Significant factors (p value<0.05) at admission associated with increased rates of mortality were altered sensorium (33.8%), respiratory rate> = 30/min (26.9%), body

temperature >102˚F (30.3%), oxygen saturation <90% (33.9%), anemia (16.7%) or blood urea nitrogen >19mg/dl (21.3%). Mortality rates were significantly higher among those receiving ICU care (31.0%), mechanical ventilation (47.8%), antiviral (29.7%) and those admitted in private hospitals (17.4%) (Table 4). Multivariable analysis showed that those who had diabetes (adj HR1.72; 95% CI 1.18–2.51), anemia (adj HR 2.68; 95% CI 1.48–4.86), oxygen saturation <90% (adj HR 4.05; 95% CI 2.45–6.69), ICU admission (adj HR 2.82; 1.71–4.66) and mechanical ventilation (adj HR 2.82; 95% CI 1.71–4.66) were at higher risk of mortality. Private hospital admission was associated with a lower risk of mortality (adj HR 0.49; 95% CI 0.3–0.8) and antiviral use was not associated with mortality after adjusting for baseline characteristics, condition at time of admission, influenza detection, and care received. (Table 5).

A sensitivity analysis was done by excluding the data from private tertiary hospital in Kolkata which reported high ICU admission and mortality. Among these pneumonia cases, ICU admission was observed in 27.6% (251/908) and death in hospital or 30-day post discharge was observed in 7.6% (69/908) (Table not shown).

## Discussion

Our study shows that four in ten cases of older adults hospitalized with CAP received ICU care and 15% of participants succumbed either in-hospital or within 30-day post discharge. At least one in seven CAP episodes were associated with influenza and RSV.

These estimates will therefore help in accomplishing the primary goal of this INSPIRE platform, which was to fill the data gaps in the burden pyramid for pneumonia and influenza among older adults aged ≥ 60 years in India [17]. Data on hospitalized non-COVID-19 pneumonia among older adults in India are limited. Only few studies from India have previously reported viral etiology of hospitalized pneumonia among older adults. The current study reported influenza positivity to vary between 9.4% to 16.9% among sites which is similar to studies from India by Hirve et al. (5.7% to 12.2%), Para et al. (13.4%), Chadha et al. (12.7%) among adults aged ≥60 years [15, 22, 23]. Influenza vaccine coverage was found to be less than 1% in this study, similar to 1.5% reported in a nationally representative study from India on vaccination among older adults during the same period [24].

A very high proportion of CAP cases were admitted to ICU in this study, though it varied widely between 8% to 73% by sites (Table 1). While studies on ICU admission among CAP hospitalized cases are limited from India, studies from developed countries have put it between 10% to 23% [25, 26]. The 30-day mortality among older adults reported in current study (15%) is higher than those reported by other similar studies– 4.6% in Switzerland, 6.8% of 30-day mortality in Canada, 5.1% in Netherlands [27–29]. It is lower than reported in Portugal (24%) and by a study from private sector hospital in India (38%) [30, 31]. About one third of the deaths happened after discharge. This could be either because of the persistent effect of the illness beyond the period of hospitalization as well as the practice of family members in India to take terminally ill patients' home for either financial or logistic reasons.

Occurrence of adverse outcomes are dependent on the patient risk-profile and management practices of a facility. For example, in this study, the tertiary private facility in Kolkata was catering to a very different clientele and reported very high ICU rate and high mortality rates. A higher use of ICU in private sector in India is anecdotally well known as a cost-recovery mechanism, especially if one is covered by health insurance. Higher use of ICU in private sector may also be attributed to more severe cases getting admitted in private hospitals as indicated by higher proportion of participants with CURB-65 score > 2 admitted in private sector hospitals. This is also substantiated by the finding that although private hospitals had higher ICU and mortality rates in univariate analyses, they had lower risk of mortality when adjusting

**Table 4. ICU admission and in-hospital/30 days post discharge mortality among hospitalized pneumonia cases among older adults by baseline characteristics, etiological agents, condition at time of admission and care received.**

| Variables | N | ICU admission | | In Hospital/30 days post-discharge mortality | |
|---|---|---|---|---|---|
| | 1090 | n(%) | p value | n(%) | p value |
| **Level 1: Baseline characteristics** | | | | | |
| **Age group** | | | | | |
| 60–64 | 313 | 117 (37.4) | 0.001 | 46 (14.7) | 0.04 |
| 65–74 | 486 | 169 (34.8) | | 61 (12.6) | |
| > = 75 | 291 | 143 (49.1) | | 56 (20.9) | |
| **Sex** | | | | | |
| Females | 419 | 167 (39.9) | 0.79 | 65 (15.5) | 0.68 |
| Males | 671 | 262 (39.0) | | 98 (14.6) | |
| **Current Smoking status** | | | | | |
| No | 856 | 355 (41.5) | 0.01 | 133(15.5) | 0.30 |
| Yes | 234 | 74 (31.6) | | 30 (12.8) | |
| **Chronic respiratory disease** | | | | | |
| No | 542 | 178 (32.8) | 0.001 | 69 (12.7) | 0.04 |
| Yes | 548 | 251 (45.8) | | 94 (17.1) | |
| **Diabetes** | | | | | |
| No | 785 | 281 (35.8) | 0.001 | 93 (11.8) | 0.001 |
| Yes | 305 | 148 (48.5) | | 70 (23.0) | |
| **Cardio-vascular disease or hypertension** | | | | | |
| No | 594 | 179 (30.1) | 0.001 | 60 (10.1) | 0.001 |
| Yes | 496 | 250 (50.4) | | 103 (20.7) | |
| **Level 2: Etiological agents** | | | | | |
| **Influenza** | | | | | |
| Negative | 956 | 367 (38.4) | 0.08 | 134 (14.0) | 0.02 |
| Positive | 134 | 62 (46.3) | | 29 (21.6) | |
| **RSV** | | | | | |
| Negative | 1066 | 418 (39.2) | 0.51 | 159 (14.9) | 0.77 |
| Positive | 24 | 11 (45.8) | | 4 (16.7) | |
| **Other Respiratory viruses** | | | | | |
| Negative | 1021 | 400 (39.2) | 0.6 | 156 (15.3) | 0.2 |
| Positive | 69 | 29 (42) | | 7 (10.1) | |
| **Level 3: Condition at time of admission** | | | | | |
| **Mental status** | | | | | |
| Conscious /oriented | 1019 | 370 (36.3) | 0.001 | 139 (13.6) | 0.001 |
| Confused/unconscious | 71 | 59 (83.1) | | 24 (33.8) | |
| **Respiratory rate[#]** | | | | | |
| <30/min | 984 | 355 (36.1) | 0.001 | 135 (13.6) | 0.001 |
| > = 30/min | 104 | 45 (44.1) | | 28 (26.9) | |
| **Body temperature[#]** | | | | | |
| < = 102 F | 1020 | 383 (37.2) | 0.001 | 142 (13.9) | 0.004 |
| >102 F | 66 | 53 (80.3) | | 20 (30.3) | |
| **Systolic BP<90 mmHg or Diastolic BP< = 60[#]** | | | | | |
| No | 659 | 280 (42.5) | 0.92 | 84 (12.7) | 0.48 |
| Yes | 16 | 7 (43.8) | | 3 (18.8) | |
| **Oxygen saturation[#]** | | | | | |

*(Continued)*

**Table 4.** (Continued)

| Variables | N | ICU admission | | In Hospital/30 days post-discharge mortality | |
|---|---|---|---|---|---|
| | **1090** | **n(%)** | **p value** | **n(%)** | **p value** |
| > = 94% | 694 | 164 (23.6) | 0.001 | 51 (7.3) | 0.001 |
| 93–90% | 151 | 84 (55.6) | | 29 (19.2) | |
| <90% | 245 | 181 (73.9) | | 83 (33.9) | |
| **Anemia#** | | | | | |
| No | 275 | 80 (29.1) | 0.001 | 16 (5.8) | 0.001 |
| Yes | 640 | 282 (44.1) | | 107 (16.7) | |
| **TLC#** | | | | | |
| 5-20K /dl | 732 | 285 (38.9) | 0.51 | 87 (11.9) | 0.01 |
| <5k/dl | 44 | 21 (47.7) | | 11 (25.0) | |
| >20K/dl | 128 | 51 (39.8) | | 24 (18.8) | |
| **Blood urea nitrogen#** | | | | | |
| < = 19 mg/dl | 474 | 146 (30.8) | 0.001 | 38 (8.0) | 0.001 |
| >19mg/dl | 361 | 196 (54.3) | | 77 (21.3) | |
| **Level 4: Care received and hospital type** | | | | | |
| **Admitted/ referred to ICU** | | | | | |
| No | 661 | - | | 30 (4.5) | 0.001 |
| Yes | 429 | - | | 133 (31.0) | |
| **Mechanical ventilation** | | | | | |
| No | 998 | - | | 119 (11.9) | 0.001 |
| Yes | 92 | - | | 44 (47.8) | |
| **Antiviral use** | | | | | |
| No | 841 | - | | 89 (10.6) | 0.001 |
| Yes | 249 | - | | 74 (29.7) | |
| **Type of Hospital** | | | | | |
| Public | 399 | 49 (12.3) | 0.001 | 43 (10.8) | 0.003 |
| Private | 691 | 380 (55.0) | | 120 (17.4) | |

for comorbid conditions, severity at time of admission, use of mechanical ventilation and antivirals.

In adjusted analysis, outcomes of influenza associated pneumonia were not different from other pneumonias. Similar results have also been reported in Netherlands by Spoorenberg et al. [29]. Clinical factors like altered sensorium, respiratory rate > = 30 per minute, high blood urea nitrogen, anemia, comorbid conditions like diabetes and SpO2 <90% were associated with ICU admission and mortality. These findings are consistent with known literature and corroborate the criteria used for categorizing patients needing hospitalization as per government of India guidelines [32–35]. These findings support the seasonal influenza triaging criteria for treatment followed by Government of India which include hemoptysis, seizures, decreased urine output, tachypnea, SpO2 <90% and hypotension [36]. With wide availability of pulse-oximeters in primary care following COVID-19, pulse-oximetry findings along with clinical scoring systems like CURB-65 can be used for assessment and prompt referral of severe pneumonia cases.

The findings from this study highlight the need for policy considerations for vaccination of older adults against pneumonia causing pathogens like influenza and RSV. Influenza vaccination has been observed to reduce hospitalization by 37% and pneumonia associated mortality by 70% in the vaccinated older adults (≥65 years) during influenza season [37, 38].

**Table 5. Factors associated with mortality during hospitalization or within 30-day post discharge among older adults with pneumonia in India 2018–2020.**

| Factors | Model 1 HR (95% CI) | Model2 HR (95% CI) | Model 3 HR (95% CI) | Model 4 HR (95% CI) | Model 5 HR (95% CI) | Model 6 HR (95% CI) | Model 7 HR (95% CI) |
|---|---|---|---|---|---|---|---|
| Cardiovascular disease or hypertension | 1.79 (1.28–2.53) | | | | 1.89 (1.36–2.63) | 1.64 (1.09–2.49) | 1.41 (0.93–2.16) |
| Diabetes | 1.69 (1.22–2.34) | | | | 1.62 (1.17–2.24) | 1.84 (1.26–2.68) | 1.72 (1.18–2.51) |
| Influenza | | 1.64 (1.09–2.44) | | | 1.49 (0.99–2.23) | 1.08 (0.65–1.8) | 0.95 (0.56–1.59) |
| Blood urea nitrogen >19gm/dl | | | 1.82 (1.01–3.27) | | | 1.46 (0.97–2.21) | 1.42 (0.94–2.14) |
| Confused/unconscious | | | 3.42 (1.67–6.98) | | | 3.04 (1.8–5.11) | 0.9 (0.47–1.73) |
| Anemia | | | 3.33 (1.41–7.83) | | | 2.43 (1.37–4.31) | 2.68 (1.48–4.86) |
| Oxygen saturation | | | | | | | |
| 90–93% | | | 3.02 (1.51–6.06) | | | 2.22 (1.26–3.89) | 2.01 (1.14–3.55) |
| <90% | | | 4.93 (2.49–9.74) | | | 5.89 (3.75–9.24) | 4.05 (2.45–6.69) |
| ICU admission | | | | 7.16 (4.49–11.42) | | | 3.31 (1.75–6.28) |
| Mechanical ventilation | | | | 1.51 (1.03–2.22) | | | 2.82 (1.71–4.66) |
| Antiviral use | | | | 1.56 (1.11–2.18) | | | 1.29 (0.84–1.96) |
| Private sector hospitalization | | | | 0.62 (0.42–0.92) | | | 0.49 (0.3–0.8) |

Model 1 include level 1 factors (Age, sex, cardiovascular disease, diabetes, chronic respiratory disease, current smoking status); Model 2 include level 2 factors (Influenza, RSV);Model 3 included level 3 factors (Mental Status, respiratory rate, body temperature, oxygen saturation, blood urea nitrogen, total leukocyte count, hemoglobin level);Model 4 included level 4 factors (ICU care, mechanical ventilation, antiviral use, type of hospital);Model 5 included cardio vascular diseases, diabetes, and influenza; Model 6 included cardio vascular diseases, diabetes, influenza, blood urea nitrogen level, mental status, anemia and oxygen saturation; Model 7 included cardio vascular diseases, diabetes, influenza, blood urea nitrogen level, mental status, anemia, oxygen saturation, ICU care, mechanical ventilation, antiviral use and type of hospital.

Immunogenicity of pneumococcal vaccine and its importance in prevention of pneumonia in older adults has been highlighted by previous studies from India [39, 40]. The needs of studies to evaluate the cost-benefit of vaccines introduction for elderly had also been previously advocated [41, 42].

Our study is one of the very few studies among adults aged ≥ 60 years hospitalized with CAP in LMIC setting. The strength of this study is that this was a multicentric study from 16 health facilities in four sites from different parts of India with a public-private and secondary-tertiary mix, and large sample size. We also had very little loss to follow-up (8% at 30-day post discharge). We also tested for influenza and RSV using qPCR in pre-COVID-19 period which could serve as a good comparator for future studies with co-circulation of SARS-CoV2.

Nevertheless, our convenience sample of older adults enrolled at the four study sites may not be representative of hospitalizations in India. All the four sites in the study were in urban area, but that reflects the fact that most secondary and tertiary health facilities in India are in urban areas. We did not have complete clinical information in almost half of the participants, but this is also reflective of lack of standardized approach to clinical management in Indian facilities as well as poor documentation. We could collect only upper respiratory specimens, and therefore could not test for bacterial agents which was another limitation of this study.

Despite these limitations, our study findings highlight the need for standardizing diagnosis and clinical management of older adults aged ≥ 60 years hospitalized with CAP especially in resource limited settings. These include guidelines for initiation of anti-viral treatment against influenza within 48 hours of onset even among non-hospitalized high-risk patient presenting with suspected influenza [43]. The current guidelines in India for seasonal influenza [33] need revision in view of the availability of newer data from epidemiological investigations and therapeutic trials globally [44]. The need for updating the clinical management guidelines has assumed more importance in the post-COVID-19 period with co-circulation of SARS-CoV2 along with influenza, RSV, and other respiratory pathogens and availability of newer vaccines against this pathogens [45, 46]. We hope our study will be useful for development of comprehensive guidelines for clinical management of CAP among older adults in resource limited settings.

## Acknowledgments

We acknowledge the support of all the investigators and coinvestigators, our research staff, study participants and other members of the community who helped us in conducting this study. Comments received from the CDC e-clearance system to improve the manuscript are acknowledged.

## Author Contributions

**Conceptualization:** Suman Kanungo, Aslesh O. Prabhakaran, Krishna Sarda, Siddhartha Saha, Anand Krishnan.

**Data curation:** Suman Kanungo, Aslesh O. Prabhakaran, Rakesh Kumar.

**Formal analysis:** Uttaran Bhattacharjee, Rakesh Kumar, Byomkesh Manna.

**Investigation:** Prabu Rajkumar, Sumit Dutt Bhardwaj, Alok Kumar Chakrabarti, Girish Kumar C. P., Varsha Potdar.

**Methodology:** Suman Kanungo, Aslesh O. Prabhakaran, Rakesh Kumar, Byomkesh Manna, Ritvik Amarchand, Avinash Choudekar, Giridara Gopal, Krishna Sarda, Siddhartha Saha, Anand Krishnan.

**Supervision:** Suman Kanungo, Prabu Rajkumar, Alok Kumar Chakrabarti, Girish Kumar C. P., Varsha Potdar, Avinash Choudekar, Krishna Sarda, Siddhartha Saha, Lalit Dar, Anand Krishnan.

**Writing – original draft:** Suman Kanungo, Uttaran Bhattacharjee.

**Writing – review & editing:** Aslesh O. Prabhakaran, Rakesh Kumar, Prabu Rajkumar, Sumit Dutt Bhardwaj, Alok Kumar Chakrabarti, Girish Kumar C. P., Varsha Potdar, Byomkesh Manna, Ritvik Amarchand, Avinash Choudekar, Giridara Gopal, Kathryn E. Lafond, Eduardo Azziz-Baumgartner, Siddhartha Saha, Lalit Dar, Anand Krishnan.

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
