## [Decision Letter · Decision Letter 0]

6 Dec 2023

PONE-D-23-32246Outcomes among adults aged 60 years and more hospitalized with pneumonia and their determinants: a prospective multi-centric hospital-based study in IndiaPLOS ONE

Dear Dr. Kanungo,

Thank you for submitting your manuscript to PLOS ONE. After careful consideration, we feel that it has merit but does not fully meet PLOS ONE’s publication criteria as it currently stands. Therefore, we invite you to submit a revised version of the manuscript that addresses the points raised during the review process.

We look forward to receiving your revised manuscript.

Kind regards,

Nosheen Nasir

Academic Editor

PLOS ONE

 [The study was funded by the Centers for Disease Control and Prevention, Atlanta, USA under Co-operative Agreement U01IP001074.].  

[None] 

6. our ethics statement should only appear in the Methods section of your manuscript. If your ethics statement is written in any section besides the Methods, please move it to the Methods section and delete it from any other section. Please ensure that your ethics statement is included in your manuscript, as the ethics statement entered into the online submission form will not be published alongside your manuscript. 

7. Please include your tables as part of your main manuscript and remove the individual files. Please note that supplementary tables (should remain/ be uploaded) as separate ""supporting information"" files".

Reviewers' comments:

Reviewer's Responses to Questions

**Comments to the Author**

1. Is the manuscript technically sound, and do the data support the conclusions?

Reviewer #1: Yes

Reviewer #2: Yes

2. Has the statistical analysis been performed appropriately and rigorously? 

Reviewer #1: Yes

Reviewer #2: Yes

3. Have the authors made all data underlying the findings in their manuscript fully available?

Reviewer #1: Yes

Reviewer #2: Yes

4. Is the manuscript presented in an intelligible fashion and written in standard English?

Reviewer #1: Yes

Reviewer #2: Yes

5. Review Comments to the Author

Reviewer #1: Community acquired pneumonia (CAP) is a significant cause of morbidity among older adults. However, studies on the etiology & risk factors associated with adverse clinical outcomes mostly emerge from high income countries. This study attempts to identify the causes of CAP and to link them to clinical outcomes in a middle income country- India, with the ultimate goal of informing policy around managing the CAP associated burden on the medical system. Overall, the study is well designed, analysis is rigorous and conclusions are proportional to the results. Studies of this kind tend to be descriptive in nature. Nevertheless, I appreciate the authors’ attempts to provide some policy recommendations based on their data.

Reviewer #2: This article summarized the results from a multi-center study focused on 60 years or older patients and tried to estimate the adverse outcomes upon admission with CAP. Overall the authors’ goals were achieved through rigorous statistical analysis. I think upon minor revision by addressing the following issues, this manuscript can be published in this journal.

- Firstly, the title of the manuscript is confusing because it does not clearly express the idea/conclusion/goal of the article. I strongly suggest editing/paraphrasing the first part of the title (before the colon mark). A prospective title could be: “Adverse outcomes in patients hospitalized with pneumonia at age 60 or more: a prospective multi-centric hospital-based study in India”

- Introduction line 72 needs the reference that it is talking about.

- Line 76-77: “CAP hospitalization and mortality in middle income countries, however, can be dramatically higher”- needs a reference, or a reason if it is a mere speculation.

- Line 151: CRUB-65 scoring is mentioned twice. Please correct it. Also, please define “CRUB-65 scoring” briefly in the method section.

- The writing format of ‘P value’ is inconsistent. Also, I suggest not to use a dash (-) between ‘p value’ and the number. For example: line 184, 201, 205, etc. Please revise to make it consistent.

- There are only figures in the document, whereas authors can make easily make more figures to represent some important data. For example: the type of virus detected in four different sites can be presented as individual “pie charts”, which is a key outstanding information included in table 1. Similarly, the data shown in Table 2 can also be presented as pie chart for each type of facilities. Figures are always helpful for readers to get the message and compare among different sites.

6. PLOS authors have the option to publish the peer review history of their article (what does this mean?). If published, this will include your full peer review and any attached files.

Reviewer #1: No

Reviewer #2: No

---

## [Author Response · Author response to Decision Letter 0]

19 Dec 2023

Authors Responses to Reviewer Comments:

We express our acknowledgement to the reviewers for their precious assessment of our work. Subsequently, we address their concerns in a point-by-point manner.

Authors Responses to Journal requirements:

We express our acknowledgement to the editor for precious assessment of our work. Subsequently, we address his/her concerns in a point-by-point manner.

Journal requirements 1

The PLOS ONE style templates can be found at 

Response

We appreciate your feedback and want to assure you that we have made revisions to our manuscript in accordance with the style guidelines provided by PLOS ONE.

Journal requirements 2

Response

Dear Emily Chenette Editor in Chief of PLOS ONE, and Iain Hrynaszkiewicz, Director of Open Research Solutions at PLOS,

Thank you for bringing to our attention the significant citation advantage associated with depositing data in a repository, as highlighted in the article (https://doi.org/10.1371/journal.pone.0230416). 

We appreciate the valuable insights provided by PLOS ONE and the commitment to fostering open research practices. Upon reviewing your note, we acknowledge the potential benefits of depositing raw data in a repository, including the up to 25% citation advantage and the opportunity to receive an Accessible Data icon on our published paper through participating repositories (https://plos.org/open-science/open-data/#accessible-data). In light of this, we will carefully consider depositing our data in an appropriate repository to ensure our work reaches the widest possible audience. Thank you once again for your proactive efforts in promoting open research, and we look forward to actively participating in this initiative. 

Reviewer 1:

Comment:

Community acquired pneumonia (CAP) is a significant cause of morbidity among older adults. However, studies on the etiology & risk factors associated with adverse clinical outcomes mostly emerge from high income countries. This study attempts to identify the causes of CAP and to link them to clinical outcomes in a middle-income country- India, with the ultimate goal of informing policy around managing the CAP associated burden on the medical system. Overall, the study is well-designed, analysis is rigorous and conclusions are proportional to the results. Studies of this kind tend to be descriptive in nature. Nevertheless, I appreciate the authors’ attempts to provide some policy recommendations based on their data.

Response

We express our gratitude for your constructive and motivating feedback.

Reviewer 2

Comment 1:

The title of the manuscript is confusing because it does not clearly express the idea/conclusion/goal of the article. I strongly suggest editing/paraphrasing the first part of the title (before the colon mark). A prospective title could be: “Adverse outcomes in patients hospitalized with pneumonia at age 60 or more: a prospective multi-centric hospital-based study in India.

Response: We thank the reviewer for bringing this to our attention. The mentioned section has been appropriately revised in accordance with the reviewer's feedback. Therefore, the title updated in the revised manuscript as “Adverse outcomes in patients hospitalized with pneumonia at age 60 or more: a prospective multi-centric hospital-based study in India”. 

Comment 2: Section- Introduction

Introduction line 72 needs the reference that it is talking about.

Response: We express gratitude to the reviewer for bringing this to our attention, and we have now incorporated the relevant reference in the revised manuscript at line 73, that is Ref no. 6: Lafond KE, Porter RM, Whaley MJ, Suizan Z, Ran Z, Aleem MA, et al. Global burden of influenza-associated lower respiratory tract infections and hospitalizations among adults: A systematic review and meta-analysis. PLoS Med. 2021;18: 1–17. doi:10.1371/JOURNAL.PMED.1003550

Comment 3: Section- Introduction

Line 76-77: “CAP hospitalization and mortality in middle income countries, however, can be dramatically higher”- needs a reference, or a reason if it is a mere speculation.

Response: We have addressed the reviewer's concern by including the relevant references into lines 77-78 of the revised manuscript. The relevant ref nos. are Ref no. 9 and Ref no.10. 

Ref no. 9: Austin S, Murthy S, Wunsch H, Adhikari NKJ, Karir V, Rowan K, et al. Access to urban acute care services in high- vs. middle-income countries: An analysis of seven cities. Intensive Care Med. 2014;40: 342–352. doi:10.1007/S00134-013-3174-7/TABLES/5. 

Ref no. 10: Salluh JIF, Kawano-Dourado L. Implementing the severe community-acquired pneumonia guidelines in low- and middle-income countries. Intensive Care Med. 2023;49: 1392–1396. doi:10.1007/S00134-023-07220-7/METRICS. 

Comment 4: Section- Introduction and method

Line 151: CRUB-65 scoring is mentioned twice. Please correct it. Also, please define “CRUB-65 scoring” briefly in the method section.

Response: We have addressed the reviewer's concern and corrected it accordingly in line 152 of the revised manuscript. Furthermore, we have addressed the reviewer's concern on CURB-65 scoring and briefly defined the "CRUB-65 scoring" in the method section in lines 152-155.

Comment 4: 

The writing format of ‘P value’ is inconsistent. Also, I suggest not to use a dash (-) between ‘p value’ and the number. For example: line 184, 201, 205, etc. Please revise to make it consistent.

Response: Thank you for your valuable feedback. We accept the inconsistency in the writing format of 'P value'. We revised the manuscript to ensure a consistent format for 'p value' without the use of a dash as per the reviewer’s recommendation.

Comment 5:

There are only figures in the document, whereas authors can make easily make more figures to represent some important data. For example: the type of virus detected in four different sites can be presented as individual “pie charts”, which is a key outstanding information included in table 1. Similarly, the data shown in Table 2 can also be presented as pie chart for each type of facilities. Figures are always helpful for readers to get the message and compare among different sites.

Response:

We appreciate the time and effort you dedicated to reviewing our manuscript. Your suggestion to represent the type of virus detected in four different sites as individual "pie charts" and to present the data in Table 2 as pie charts for each type of facility is really thoughtful and may conveying information effectively to readers.

However, we would like to clarify that we try to provide a comprehensive overview and we intentionally opted to present all relevant data in a single consolidated table rather than dispersing it multiple figures. Our aim was to represent the interrelationships among different variables.

Authors Responses to Journal requirements:

We express our acknowledgement to the editor for precious assessment of our work. Subsequently, we address his/her concerns in a point-by-point manner.

Journal requirements 1

The PLOS ONE style templates can be found at 

Response

We appreciate your feedback and want to assure you that we have made revisions to our manuscript in accordance with the style guidelines provided by PLOS ONE.

Journal requirements 2

Response

Dear Emily Chenette Editor in Chief of PLOS ONE, and Iain Hrynaszkiewicz, Director of Open Research Solutions at PLOS,

Thank you for bringing to our attention the significant citation advantage associated with depositing data in a repository, as highlighted in the article (https://doi.org/10.1371/journal.pone.0230416). 

We appreciate the valuable insights provided by PLOS ONE and the commitment to fostering open research practices. Upon reviewing your note, we acknowledge the potential benefits of depositing raw data in a repository, including the up to 25% citation advantage and the opportunity to receive an Accessible Data icon on our published paper through participating repositories (https://plos.org/open-science/open-data/#accessible-data).

In light of this, we will carefully consider depositing our data in an appropriate repository to ensure our work reaches the widest possible audience. Thank you once again for your proactive efforts in promoting open research, and we look forward to actively participating in this initiative. 

Journal requirements 3

Thank you for stating the following financial disclosure: 

 [The study was funded by the Centers for Disease Control and Prevention, Atlanta, USA under Co-operative Agreement U01IP001074.]. 

Response 

We maintained the role of funders in cover letter for your kind reference. 

Journal requirements 4

Thank you for stating the following in your Competing Interests section: 

[None] 

Response 

Thank you for your prompt attention to our Competing Interests section. We rewrite the Competing Interests section in the revised manuscript and in cover letter also as “All the authors have declared that no competing interests exist”. 

Journal requirements 5

Please amend either the abstract on the online submission form (via Edit Submission) or the abstract in the manuscript so that they are identical.

Response 

We ensure that the abstract on the online submission form via the Edit Submission feature are perfectly aligns with the abstract in the manuscript.

Journal requirements 6

our ethics statement should only appear in the Methods section of your manuscript. If your ethics statement is written in any section besides the Methods, please move it to the Methods section and delete it from any other section. Please ensure that your ethics statement is included in your manuscript, as the ethics statement entered into the online submission form will not be published alongside your manuscript.

Response

Thank you for your prompt attention and we include the ethics statement in methods section of our revised manuscript.

Journal requirements 7

Please include your tables as part of your main manuscript and remove the individual files. Please note that supplementary tables (should remain/ be uploaded) as separate ""supporting information"" files".

Response

We include our tables as part of our main revised manuscript and remove the individual files. 

Journal requirements 8

Response

We review our reference list and ensure that all the reference are correct and complete.

---

## [Decision Letter · Decision Letter 1]

4 Jan 2024

Adverse outcomes in patients hospitalized with pneumonia at age 60 or more: a prospective multi-centric hospital-based study in India

PONE-D-23-32246R1

Dear Dr. Kanungo,

We’re pleased to inform you that your manuscript has been judged scientifically suitable for publication and will be formally accepted for publication once it meets all outstanding technical requirements.

Kind regards,

Nosheen Nasir

Academic Editor

PLOS ONE

Additional Editor Comments (optional):

Reviewers' comments:

Reviewer's Responses to Questions

**Comments to the Author**

1. If the authors have adequately addressed your comments raised in a previous round of review and you feel that this manuscript is now acceptable for publication, you may indicate that here to bypass the “Comments to the Author” section, enter your conflict of interest statement in the “Confidential to Editor” section, and submit your "Accept" recommendation.

Reviewer #2: All comments have been addressed

2. Is the manuscript technically sound, and do the data support the conclusions?

Reviewer #2: Yes

3. Has the statistical analysis been performed appropriately and rigorously? 

Reviewer #2: Yes

4. Have the authors made all data underlying the findings in their manuscript fully available?

Reviewer #2: (No Response)

5. Is the manuscript presented in an intelligible fashion and written in standard English?

Reviewer #2: Yes

6. Review Comments to the Author

Reviewer #2: Thanks to the authors for updating the manuscript. I hope it will add value to the epidemiological research area and be helpful for informed decision making by the health-care system.

7. PLOS authors have the option to publish the peer review history of their article (what does this mean?). If published, this will include your full peer review and any attached files.

Reviewer #2: No
